# Online Home-Based Physical Activity Counteracts Changes of Redox-Status Biomarkers and Fitness Profiles during Treatment Programs in Postsurgery Female Breast Cancer Patients

**DOI:** 10.3390/antiox12051138

**Published:** 2023-05-22

**Authors:** Chantalle Moulton, Elisa Grazioli, Cristina Antinozzi, Cristina Fantini, Claudia Cerulli, Arianna Murri, Guglielmo Duranti, Roberta Ceci, Maria Chiara Vulpiani, Patrizia Pellegrini, Sveva Maria Nusca, Francesco Cavaliere, Simona Fabbri, Paolo Sgrò, Luigi Di Luigi, Daniela Caporossi, Attilio Parisi, Ivan Dimauro

**Affiliations:** 1Unit of Biology and Genetics of Movement, Department of Movement, Human and Health Sciences, University of Rome Foro Italico, 00135 Rome, Italy; c.moulton@studenti.uniroma4.it (C.M.); cristina.fantini@uniroma4.it (C.F.); daniela.caporossi@uniroma4.it (D.C.); 2Unit of Physical Exercise and Sport Sciences, Department of Movement, Human and Health Sciences, University of Rome Foro Italico, 00135 Rome, Italy; elisa.grazioli@uniroma4.it (E.G.); claudia.cerulli@uniroma4.it (C.C.); a.murri@studenti.uniroma4.it (A.M.); attilio.parisi@uniroma4.it (A.P.); 3Endocrinology Unit, Department of Movement, Human and Health Sciences, University of Rome Foro Italico, 00135 Rome, Italy; cristina.antinozzi@uniroma4.it (C.A.); paolo.sgro@uniroma4.it (P.S.); luigi.diluigi@uniroma4.it (L.D.L.); 4Unit of Biochemistry and Molecular Biology, Department of Movement, Human and Health Sciences, University of Rome Foro Italico, 00135 Rome, Italy; guglielmo.duranti@uniroma4.it (G.D.); roberta.ceci@uniroma4.it (R.C.); 5Department of Medical-Surgical and Translational Medicine Sciences, La Sapienza University of Rome, 00185 Rome, Italy; mariachiara.vulpiani@gmail.com (M.C.V.); patrizia.pellegrini@uniroma1.it (P.P.); sveva.nusca@uniroma1.it (S.M.N.); 6Unit of Breast Surgery, Center of Breast of Belcolle Hospital, 01100 Viterbo, Italy; francesco.cavaliere@asl.vt.it (F.C.); simona.fabbri@asl.vi.it (S.F.)

**Keywords:** breast cancer, exercise, heat-shock proteins, oxidative stress, antioxidants, cytokines

## Abstract

Breast cancer (BC) is one of the most commonly diagnosed types of cancer in women. Oxidative stress may contribute to cancer etiology through several mechanisms. A large body of evidence indicates that physical activity (PA) has positive effects on different aspects of BC evolution, including mitigation of negative effects induced by medical treatment. With the aim to verify the capacity of PA to counteract negative effects of BC treatment on systemic redox homeostasis in postsurgery female BC patients, we have examined the modulation of circulating levels of oxidative stress and inflammation markers. Moreover, we evaluated the impacts on physical fitness and mental well-being by measuring functional parameters, body mass index, body composition, health-related quality of life (QoL), and fatigue. Our investigation revealed that PA was effective in maintaining plasma levels of superoxide dismutase (SOD) activity and tGSH, as well as peripheral blood mononuclear cells’ (PBMCs) mRNA levels of SOD1 and heat-shock protein 27. Moreover, we found a significant decrease in plasma interleukin-6 (≈0.57 ± 0.23-fold change, *p* < 0.05) and increases in both interleukin-10 (≈1.15 ± 0.35-fold change, *p* < 0.05) and PBMCs’ mRNA level of SOD2 (≈1.87 ± 0.36-fold change, *p* < 0.05). Finally, PA improves functional parameters (6 min walking test, ≈+6.50%, *p* < 0.01; Borg, ≈−58.18%, *p* < 0.01; sit-and-reach, ≈+250.00%, *p* < 0.01; scratch right, ≈−24.12%, and left, ≈−18.81%, *p* < 0.01) and body composition (free fat mass, ≈+2.80%, *p* < 0.05; fat mass, ≈−6.93%, *p* < 0.05) as well as the QoL (physical function, ≈+5.78%, *p* < 0.05) and fatigue (cognitive fatigue, ≈−60%, *p* < 0.05) parameters. These results suggest that a specific PA program not only is effective in improving functional and anthropometric parameters but may also activate cellular responses through a multitude of actions in postsurgery BC patients undergoing adjuvant therapy. These may include modulation of gene expression and protein activity and impacting several signaling pathways/biological activities involved in tumor-cell growth; metastasis; and inflammation, as well as moderating distress symptoms known to negatively affect QoL.

## 1. Introduction

According to the World Health Organization (WHO), cancer is one of the main causes of mortality worldwide [1]. Particularly, female breast cancer (BC) has surpassed lung cancer as the most commonly diagnosed cancer, with an estimated 2.3 million new cases (11.7%) [1].

Systemic oxidative stress has been implicated in many diseases and disorders, including the pathogenesis and progression of BC [2,3,4]. Moreover, it is known that most anticancer drugs act through oxidative-stress pathways via production of reactive oxygen species (ROSs) to kill cancer cells, which, as a common side effect, induces oxidative-stress tissue toxicity (e.g., cardiotoxicity) [5,6]. Oxidative stress refers to excessive production of ROSs, particularly from oxygen radicals, during a period of increased exposure to environmental stress. ROSs are a normal byproduct of metabolism and are necessary components of both cell signaling and homeostasis [7]; however, excessive amounts can reduce the antioxidant defenses of the body, resulting in negative effects on health. ROSs can induce damage to lipids, proteins, and DNA, which in turn lead to genetic mutations and genomic instability, thus contributing to carcinogenesis [2,8]. These aspects make ROSs a potential target for cancer therapies.

Over the past decades, physical activity (PA) has emerged as an integrative therapy to improve cancer survival outcomes, and international guidelines include regular PA as a recommendation for patients [9]. In women with BC, PA during adjuvant therapy has been shown to reduce loss of muscle strength [10], improve health-related quality of life (QoL), and reduce cancer-related fatigue as well as other negative side effects of treatments [11,12,13]. Some of these improvements may be related to the capacity of PA to counteract the negative effects of BC treatments on systemic redox homeostasis. Despite the documented benefits of PA, a decrease in PA is very often observed after BC diagnosis, and a large proportion of BC survivors do not reach the recommended levels of PA, suggesting that targeted intervention is necessary within this population [14].

Recently, with the aim of monitoring and improving fitness- and health-related behaviors in patients with different diseases, several studies have highlighted the effects of e-health-based protocols, developed using Internet resources, for monitoring and connecting these people [15,16]. To date, this new approach has been widely applied in different healthcare areas to improve communication between patients and specialist doctors, to monitor patients’ activities, and to give electronic access to diagnosis or screening results, especially in the oncology field [17,18,19,20].

Although there is evidence in BC survivors regarding the positive effects of exercise protocol performed at home, using video lessons and supervised by a specialized trainer through video-calling, on QoL, fatigue, aerobic fitness, body mass, and muscular strength, none of these studies have yet evaluated the impact of PA on markers of redox homeostasis and inflammation [19,21,22,23,24].

On these bases, the aim of this pilot study was to improve the knowledge in the field of “exercise oncology” and particularly on the effects of PA on plasmatic markers of oxidative stress and inflammation in postsurgery women, with BC, undergoing adjuvant therapy (i.e., chemotherapy, hormone therapy, and radiotherapy). In these patients, we investigated the effects of a 16-week online exercise program on modulation of superoxide dismutase (SOD) and catalase (CAT) activity, total glutathione (tGSH), lipid oxidation (TBARs), total antioxidant capacity (TAC), and total free thiols (tFTHs), as well as interleukin-6 (IL6), interleukin-8 (IL8), interleukin-10 (IL10), and tumor necrosis factor alpha (TNFα), in postsurgery female BC patients undergoing adjuvant therapy (i.e., chemotherapy, hormone therapy, and radiotherapy). To assess possible exercise-induced modulatory effects, we also evaluated, in peripheral blood mononuclear cells (PBMCs), the gene expressions of the most exercise-responsive molecules, such as heat-shock proteins 70 and 27 (HSP70 and HSP27), superoxide dismutase 1 and 2 (SOD1 and SOD2), and glutathione peroxidase 1 (GPx1). Finally, to quantify the effects of this exercise program on physical fitness and mental well-being, we also analyzed different functional parameters (i.e., balance, flexibility, strength, and functional capacity), QoL, body mass index (BMI), and body composition.

## 2. Materials and Methods

### 2.1. Study Design

Female volunteers affected by breast cancer (*n* = 20, 45–65 years) were enrolled at the Sant’Andrea Hospital/UOC of Medical Oncology in Rome (Italy) and Belcolle Hospital/UOC of Breast Surgery in Viterbo (Italy). Eligibility criteria included women diagnosed with stage I-III breast cancer; >18 years old; literate in Italian; and scheduled to undergo adjuvant therapy (radiotherapy, hormone therapy, chemotherapy). Exclusion criteria included being unable to perform basic activities of daily living; cognitive disorders or severe emotional instability; other disabling comorbidities that might hamper physical training (e.g., severe heart failure, chronic obstructive pulmonary disease, orthopedic conditions, and neurological disorders); being active ≥150 min/wk 3 months prior to surgical treatment; and undergoing neoadjuvant therapy.

After inclusion, all volunteers underwent detailed medical examination, including anamnesis, physical examination, electrocardiogram at rest (ECG), and resting heart rate after 15 min of bed rest, and completed a detailed eating-habit diary in which all foods and drinks consumed during the 3 consecutive days before beginning the training protocol were recorded. Moreover, for each volunteer, body composition analysis (DS Medical’s Handy 3000) and anthropometric measures (i.e., height, weight, and BMI) were collected at the baseline visit and at the end of the experimental period.

To analyze the impact of exercise training, the volunteers were randomly assigned to two different groups: the Control Group (CG, *n* = 10), where all subjects received the usual cancer treatments from the hospital, which included chemotherapy and/or hormone therapy and/or radiotherapy; and the Exercise Group (EG, *n* = 10), where the participants were additionally included in an exercise training program.

This pilot study was approved by the Ethics Committee of the University of Rome “Sapienza” (RIF.CE: 5451_2019), and all subjects provided informed consent. This study was conducted in accordance with the Declaration of Helsinki.

### 2.2. Randomization and Blinding Procedures

A computer-generated list of random numbers was used to allocate the study volunteers to one of the study groups, either the CG or the EG, according to simple randomization procedures. Sequentially numbered opaque envelopes were used for concealed randomization. These envelopes contained group assignments and were provided to the participants after the baseline assessment. All data of the participants who did not complete the study protocol were excluded from the statistical analysis. While our study was not fully blinded, both the patients and the kinesiologist were blinded to the group allocation before the first detailed medical examination, and all people who participated in the analysis of biological samples and data remained blinded at all times.

### 2.3. Exercise Training Program

The online exercise training program was performed twice per week (1 h session), during 2 nonconsecutive days/week, for 16 weeks (32 sessions in total). A specialized kinesiologist from the University of Rome “Foro Italico” guided exercise sessions online using the platform Microsoft Teams. Training sessions were tailored to individuals and their specific fitness and fatigue levels.

Each training session consisted of four main phases: warm-up, resistance exercises (circuit resistance training using small weights), aerobic training (aerobic steps with music at ≈70% of individual heart rate reserve), and, finally, a cool-down phase.

Phase 1: a warm-up phase lasting 10 min, which included mobility exercises for the neck and shoulders as well as trunk rotations, flexions, and extensions. Additionally, balance exercises were included in this phase.

Phase 2: a resistance exercise phase lasting from 20 to 40 min was implemented using the circuit-training method. Each circuit included one exercise for each of different muscle groups: upper limbs, core, back, and lower limbs, respectively. Dumbbells with different sets of weight (1 kg, 1.5 kg, or 2 kg) were used in this phase. The amount of weight for each participant was defined during the baseline functional-assessment tests; participants were asked to perform 20 reps of dumbbell bicep curls with a set weight and to rate the level of perceived fatigue with a Borg scale.

If the rating was: (1) ≤2, the weight was increased by 0.5 kg; (2) between 2 and 4, the weight was kept the same; and (3) ≥4, the weight was decreased by 0.5 kg.

Exercise volume and intensity (reps and sets) were increased every month, and recovery time was gradually decreased over time.

Phase 3: an aerobic training phase lasting up to 20 min was implemented using Tabata methodology, with 5 different exercises (lunge step, V-step, leg curl, knee-up, and kick). During weeks 1–2, this phase lasted for 10 min and was performed with a work-to-rest ratio of 40:20 s; during weeks 3–4, the same exercises were performed continuously at moderate intensity for 10–15 min. From week 4 onwards, 120 bpm music was introduced during training, and training intensity was gradually increased to reach 20 min of continuous aerobic training. Moreover, training intensity was systematically increased during the weeks, starting from 55–60% of the max heart rate (HRmax) during weeks 1–4, 60–70% of the HRmax during weeks 5–8, and 70–80% of the HRmax for the remaining period. Target heart rate (beats/min) was calculated using the Karvonen formula during medical screening and constantly monitored during training with a smartwatch (Mi Band 5) provided by the University of Rome “Foro Italico”.

Notably, some patients received COVID-19 vaccine injections during the study period, and abnormal frequencies of heart rhythm were observed in the week after each vaccination. Accordingly, training intensity was maintained at 60% of the HRmax during this period for safety reasons.

Phase 4: a cool-down phase lasting 10 min was performed with stretching exercises to improve patients’ flexibility and posture and restore the baseline heart rate (HR) after the aerobic session.

A printed copy of the 0–10 Borg scale was provided to each patient and was used to monitor the rates of perceived fatigue and difficulty during the entire training period.

### 2.4. Blood Sampling and Isolation of Peripheral Blood Mononuclear Cells (PBMCs)

Before (PRE) and after (POST) the experimental period, fasting blood samples were drawn from the antecubital vein while each subject remained in a reclined position (Figure 1). Blood sampled in EDTA tubes (BD Biosciences, Franklin Lakes, NJ, USA) was used for plasma collection by centrifugation of whole blood (2500 rpm × 10 min at 4 °C) and for PBMC isolation. Human PBMCs were purified from whole blood with Ficoll gradient (Sigma-Aldrich, Milan, Italy), as already described [25]. Plasma and PBMC samples were aliquoted and stored at −80 °C for further analyses.

### 2.5. RNA Extraction and RT-qPCR

The total RNA from the PBMCs was obtained from cells using TRIZOL (Invitrogen, Carlsbad, CA, USA), according to the manufacturer’s procedure. A real-time quantitative RT−PCR was performed on a 7500 Real Time PCR System (Applied Biosystems, Life Technologies, Waltham, MA, USA). Each 20 μL reaction mixture contained 10 μL of Power SYBR Green RNA−to Ct 1stepMaster mix (2×) (Applied Biosystems, Life Technologies, Waltham, MA, USA), 10 pmol of specific primer sets, 0.16 μL of RT Enzyme Mix (Applied Biosystems, Life Technologies, Waltham, MA, USA), and 10–15 ng of RNA samples. The RT−PCR amplification profile was as follows: the RT step at 48 °C for 30 min, followed by enzyme activation at 95 °C for 10 min and then 40 cycles of denaturation at 95 °C for 15 s and annealing/extension at 60 °C for 1min. All samples were run in triplicate. Normalization was performed utilizing cyclophilin A. A threshold cycle (CT) was observed in the exponential phase of amplification, and quantification of relative expression levels was performed with standard curves for target genes and the endogenous control. Geometric means were used to calculate the ΔΔCT (delta–delta CT) values, expressed as 2^−ΔΔCT^. The value of each control sample was set at 1 and was used to calculate the fold changes of target genes. The list of primers utilized is reported in Appendix A. For primer design, we used Primer 3 software (http://bioinfo.ut.ee/primer3-0.4.0/, accessed on 1 May 2022).

### 2.6. Multiplex Cytokine Assay

Plasma levels of TNF-α, IL-6, IL-8, and IL-10 were assayed in patients undergoing physical exercise protocol, pre-exercise (T0), and after four months (T4) using a magnetic bead-based multiplex assay (Bio-Plex^®^ Precision Pro™ Human Cytokine Assays, Bio-Rad Laboratories, Milan, Italy) according to the manufacturer’s recommendation and as previously described [26]. Data acquisition was performed with the Bio-Plex 200 System™ (Bio-Rad Laboratories, Inc., Hercules, CA, USA), which uses Luminex fluorescent bead-based technology. Data analysis was performed with Bio-Plex Manager™ 6.0 software (Bio-Rad Laboratories, Inc., Hercules, CA, USA). Samples were run in triplicate twice.

### 2.7. Total Antioxidant Capacity

Trolox^®^ equivalent antioxidant capacity (TAC) was determined spectrophotometrically [27]. This technique uses the reactivity of antioxidant compounds found in blood plasma, relative to a 1 mM Trolox^®^ (vitamin E analog) standard. The variation in the absorbance analyzed was compared to those observed using the Trolox^®^ standard.

### 2.8. Total Glutathione

The total GSH content (tGSH) (Intracellular reduced (GSH) + oxidized (GSSG) glutathione) was quantified with a 5,5-dithio-bis-(2-nitrobenzoic acid) (DTNB)-glutathione reductase-recycling assay, as previously described [28].

### 2.9. Total Plasma Free Thiol

The free thiol (FTH) concentrations in blood plasma samples were quantified with an Ellman assay [29], according to the manufacturer’s instructions. The molar concentration of thiols was calculated from the molar absorbance of the 2-nitro-5-thiobenzoate (TNB) anion and stated in µmol-SH/g plasmatic proteins.

### 2.10. Thiobarbituric Acid-Reactive Substances (TBARs)

The TBAR level was evaluated using the TBARS (TCA method) Assay Kit (Cayman Chemical Company, Ann Arbor, MI, USA, No. 700870) with spectrophotometric analysis, according to the manufacturer’s instructions [30]. This kit measured lipid peroxidation by reacting MDA with TBA under high temperatures and acidic conditions to form an MDA–TBA adduct, after which absorbance was measured at 530–540 nm.

### 2.11. SOD Activity

Cayman’s SOD Assay kit (Cayman Chemical Company, Ann Arbor, MI, USA, No. 706002), which uses tetrazolium salt for detection of superoxide radicals produced by xanthine oxidase and hypoxanthine, was used according to the manufacturer’s instructions [31]. This assay measures all forms of SOD (Cu/Zn, Mn, and FeSOD). A single unit of SOD is defined as the amount of the enzyme that is required in order to exhibit 50% dismutation of the superoxide radical. Plasma was diluted 1:5 with sample buffer before assaying for SOD activity. Results are expressed as units/mg of protein tested.

### 2.12. CAT Activity

Cayman’s Catalase Assay Kit (Cayman Chemical Company, Ann Arbor, MI, USA, No. 707002) was used to determine catalase enzyme activity from its peroxidatic function according to the manufacturer’s instructions [31]. This method is based on the reaction of the enzyme with methanol in the presence of an optimal concentration of hydrogen peroxide. The formaldehyde that is produced is colorimetrically measured with Purpald. One unit of CAT activity is defined as the amount of enzyme that will result in the formation of 1.0 nmol of formaldehyde per minute at 25 °C. Results are expressed in nmol/min/mL.

### 2.13. Physical Activity Level

Physical activity levels were assessed for all participants at baseline, using the Italian edition of the International Physical Activity Questionnaire (IPAQ), which has been shown to be a valid and reliable technique to measure frequency, type, duration, and intensity of PA, as well as sedentary behavior [32].

### 2.14. Functional Capacity Evaluation and Patient-Reported Outcomes

The following functional capacity parameters were measured before (PRE) and after (POST) the 16-week experimental program for each participant.

(1) Six-minute walking test (6MWT) and Borg CR10 scale analysis: a supervised 6MWT was performed according to the American Thoracic Society guideline [33] to assess functional capacity. Briefly, each patient was informed about the test and that they were required to walk as fast as they could for six minutes. A 30 m-long indoor corridor setting was established. At each minute, the patient was informed orally according to guidelines. Devices continually monitored heart rate and percentage of oxygen saturation. After completion of the tests, the Borg CR10 scale was used to evaluate perceived fatigue.

(2) Muscular strength and flexibility tests: muscular strength was estimated for upper-limb handgrip strength (HG) using a dynamometer (Jamar Plus^®^, Patterson Medical Ltd., Sutton-in-Ashfield, UK); this test, representative of the general strength of the subject, measures the strength of the handheld grip by tightening the dynamometer as tightly as possible. Two attempts for both hands were recorded, and the median value was evaluated. For the lower limbs, transition movements, balance, and the risk of falling, the 30-s sit-to-stand test (STS) was used. During the test, the number of times that the patient could properly get up from a chair and sit back down in 30 s was counted. Flexibility was evaluated using two specific tests, back stretch and trunk rotation, to evaluate shoulder and trunk flexibility and mobility.

(3) The European Organization for Research and Treatment of Cancer Quality of Life Questionnaire (EORTC-QLQ-C30): the EORTC-QLQ-C30 is the most frequently used cancer-specific health-related questionnaire. It is organized into five different functional subscales (PF: physical functioning; RF: role functioning; EF: emotional functioning; CF: cognitive functioning; and SF: social functioning) and eight symptom items [34]. The questionnaire consists of a total of 28 items, and each is scored “1: None” through “4: A lot” according to the 4-category-based Likert Scoring System. Higher scores indicated worse quality of life, and vice versa.

(4) Fatigue Questionnaire EORTC QLQ-FA12: the FA12 is a new fatigue module designed to complement the EORTC QLQ-C30 and consists of 12 items, with four response categories for each item, coded with values from 1 to 4 [35]. In accordance with the scales of the EORTC QLQ-C30, the FA12 scores are transformed to the range of 0–100, with higher levels indicating greater degrees of fatigue. The FA12 comprises three subscales: physical fatigue (five items), emotional fatigue (three items), and cognitive fatigue (two items). The remaining two items serve as global indicators for interference of fatigue with daily activities and social squeal of fatigue.

### 2.15. Statistical Analysis

A statistical analysis was conducted using GraphPad Prism software 9.0 (GraphPad software, San Diego, CA, USA). The Kolmogorov–Smirnov or Shapiro–Wilk test was used to test the normality of quantitative variables. Normally distributed, continuous biological variables were analyzed using two-way ANOVA for repeated measures, with the Bonferroni correction as the post hoc test, and Student’s *t*-test. Moreover, data reported in all tables were analyzed with an unpaired/paired-samples Student *t*-test. In all cases, *p*-values ≤ 0.05 were considered significant.

## 3. Results

### 3.1. Baseline Characteristics of Subjects

The characteristics of the study population, including age, weight, height, body mass index (BMI), type of intervention and treatment, as well as physical activity level, are reported in Table 1.

Both the CG and the EG were homogenous for the type of intervention and treatments. No significant differences between the EG and the CG were observed for age (50.5 ± 5.7 yrs vs. 45.1 ± 5.5 yrs, respectively; *p* > 0.05), BMI (23.1 ± 2.5 Kg/m^2^ vs. 21.3 ± 0.5 Kg/m^2^, respectively; *p* > 0.05), or physical activity level (885.6 ± 666.0 vs. 898.3 ± 988.5 MET-min/week, respectively; *p* > 0.05).

### 3.2. Systemic Level of Antioxidants and Oxidative-Stress Parameters

At each experimental time point (PRE and POST), no significant differences were observed for catalase activity, TAC, TBARs, or tFTHs between groups or within each group (*p* > 0.05) (Figure 1A,C–E). However, SOD activity and total GSH were significantly reduced only in the CG at the end of the experimental protocol (SOD, PRE vs. POST, 92.65 ± 9.79 vs. 56.60 ± 0.12, *p* = 0.024) (total GSH, PRE vs. POST, 82.26 ± 9.09 vs. 69.62 ± 5.56, *p* = 0.001), with significantly lower levels even when compared with the EG after training (SOD, EG vs. CG, 95.39 ± 0.24 vs. 56.60 ± 0.12, *p* = 0.001) (tGSH, EG vs. CG, 80.32 ± 8.32 vs. 69.62 ± 5.56, *p* = 0.048) (Figure 1B,F).

### 3.3. Plasma Levels of Cytokines

No significant differences were observed for TNF-α levels at any experimental point in both groups (Figure 2). Compared with the PRE values, IL-6 at POST increased ≈2.04 ± 0.47-fold in the CG (*p* < 0.05) and decreased ≈0.57 ± 0.23-fold in the EG (*p* < 0.05). In both the CG and the EG, IL8 at POST showed significant decreases of ≈0.60 ± 0.20-fold (*p* < 0.01) and ≈0.72 ± 0.18-fold (*p* < 0.05), respectively, with respect to the PRE values (Figure 2). In the CG, the levels of IL10 at POST significantly decreased ≈0.74 ± 0.39-fold (*p* < 0.05), while in the EG, we observed an increase of ≈1.15 ± 0.35-fold (*p* < 0.05).

### 3.4. Gene-Expression Analysis of Antioxidants and Stress-Response Proteins in PBMCs

As shown in Figure 3, the analysis of several antioxidants revealed a beneficial effect of exercise training in terms of gene expression of SOD1, which was significantly higher in the EG than in the CG at the end of the experimental protocol (POST: CG vs. EG, 9.75 ± 0.44 vs. 16.33 ± 3.60, *p* = 0.030) (Figure 3A). Indeed, the mRNA level of SOD1 was significantly decreased within the CG at the end of the experimental protocol (PRE vs. POST, 15.79 ± 3.52 vs. 9.75 ± 0.44, *p* = 0.0052) (Figure 3A).

Interestingly, SOD2 was significantly increased within the EG (PRE vs. POST, 49.28 ± 13.72 vs. 75.28 ± 18.07, *p* = 0.0215), with values higher than in the CG at the end of the experimental protocol (POST: CG vs. EG, 50.97 ± 11.22 vs. 75.28 ± 18.07, *p* = 0.0374) (Figure 3B). No effects were found for GPx1 in either group (*p* > 0.05) (Figure 3C).

Although there was a significant group interaction (*p* = 0.037), the analysis of specific stress-response proteins highlighted no effect of exercise at any experimental point, neither on the gene expression of HSP70 nor on HSP27 (*p* > 0.05) (Figure 3D,E). Only the CG showed a significant decrease in HSP27 at the end of the experimental design (PRE vs. POST, 0.73 ± 0.12 vs. 0.52 ± 0.09, *p* = 0.023) (Figure 3E).

### 3.5. Functional Parameters

As reported in Table 2, among functional parameters, the EG improved in the 6MWT (PRE vs. POST, *p* = 0.007), perceived exertion (PRE vs. POST, *p* = 0.003), and the sit-and-reach test (PRE vs. POST, *p* = 0.004), as well as in the scratch test for both arms, right (PRE vs. POST, *p* = 0.009) and left (PRE vs. POST, *p* = 0.005).

The CG showed a significant decrease in functional parameters, including in the 6MWT (PRE vs. POST, *p* = 0.044) and in the left handgrip test (PRE vs. POST, *p* = 0.002) (Table 2).

### 3.6. Body Composition

Exercise training induced a significant reduction in fat mass, of −6.9% (PRE vs. POST, *p* = 0.043), and an increased free fat mass by +2.8% (EG: PRE vs. POST, *p* = 0.049) (Table 3). On the contrary, at the end of the experimental protocol, the CG had an increase in BMI of +6.5% (CG: PRE vs. POST *p* = 0.045) matched with an increase in fat mass of 1.3% (PRE vs. POST, *p* = 0.032) (Table 3).

### 3.7. Health-Related Quality of Life and Fatigue Perception

Data regarding health-related quality of life and fatigue were measured with the EORTC QLQ-C30 and the EORTC-FA12 (Table 4). The EG showed an increase in physical function after training (PRE vs. POST, *p* = 0.042). Moreover, the other subscales (emotional function, cognitive function, social function, and global health) registered small increases, though not significant. Data related to fatigue highlighted that exercise training had a significant impact in reducing cognitive fatigue (EG: PRE vs. POST *p* = 0.44).

In the CG, no significant differences were observed for any of the parameters from the EORTC QLQ-C30 or the fatigue items (*p* > 0.05) at the end of the experimental period (Table 4).

## 4. Discussion

To our knowledge, this pilot study is the first to evaluate, at a systemic level, changes in markers related to oxidative stress and inflammation as well as physical fitness and mental-well-being parameters in postsurgery female BC patients, during a 16-week online exercise program, at the beginnings of their individual therapeutic protocols.

Oxidative changes have been described in cancer cells when compared to normal noncancerous cells, suggesting a role for occurrence of a pro-oxidant status in malignant conditions [36]. A growing number of studies have focused on investigating the redox changes that take place in solid tumors, especially in breast cancer.

As reported from different authors, BC patients show elevated lipid peroxidation levels and disturbed antioxidant enzyme activities [37]. Circulating levels of SOD and CAT are generally higher in BC patients compared with healthy groups [38,39,40]. Others found that the antioxidant activities of SOD and GPx were elevated in BC tissue compared with healthy tissue [40,41]. These results may be explained by the fact that BC patients have higher levels of oxidative stress and, thus, the body tries to compensate by increasing the antioxidant response. Indeed, as suggested by other researchers, this could represent a natural defense mechanism to fight carcinogenesis [2,3]. Although the knowledge in this field has increased, it still remains to be clarified whether increased oxidative stress is the cause or the consequence of BC or a combination of both.

Oxidative stress also plays a critical role in cancer treatment, with cytotoxic therapies increasing oxidative damage to potentially kill tumor cells. Indeed, several anticancer strategies, currently used alone or in combination, for BC are known to induce high levels of oxidative stress [42,43,44,45]. This leads to significant depletion of endogenous antioxidant systems in cancer cells and at a systemic level, inducing greater sensitivity to cell death through redox signaling alterations [46,47,48].

In our pilot study, different biomarkers were used to assess systemic redox homeostasis, e.g., enzymatic (SOD, CAT, and GPx) and nonenzymatic (tGSH) antioxidant systems, total redox status (TAC and tFTHs), and lipid oxidative damage products (TBARs). In women, with BC, undergoing medical treatments (i.e., chemotherapy, hormone therapy, and radiotherapy), we identified significant depletions in SOD activity and tGSH. These molecules are known to protect proteins, lipids, and DNA from oxidative damage as well as participate in regeneration of other antioxidants (i.e., vitamins C and E) [49]. Although no significant effects were observed in antioxidant status or lipid peroxidation, we cannot exclude effects on these parameters caused by increases in ROSs. Interestingly, patients belonging to the CG also showed reduced gene expressions of SOD1 and HSP27 (HSPB1) in PBMCs at the end of the experimental study. Human PBMCs contain many of the functional cell types of the immune system (i.e., T and B cells, natural killers, and monocytes) and represent an important cellular model to study immune responses and drug toxicity [50]. Cells may employ several mechanisms to protect themselves from the effects of toxicants, including modulation of gene-expression profiles related to molecules normally involved in protection and adaptation under various stress conditions, including redox imbalance [51,52]. The cellular response to stress is mainly represented at the molecular level by increased synthesis of antioxidants and heat-shock proteins (HSPs), which may contribute to reduced risks of different diseases [51,52,53,54]. While antioxidants scavenge free radicals from cells and prevent or reduce damage caused by oxidation, the primary mechanisms by which HSP27 act are (1) protein folding; (2) maintaining of cell structural integrity by facilitating formation of the cytoskeletal elements, influencing their functions, and protecting the cytoskeleton during stress; (3) reduction in oxidative stress through modulation of enzymes involved in the ROS-Glutathione pathway; and (4) suppression of apoptosis or other kinds of cell death through inhibition of procaspase, caspase 3, and apoptosis signal-regulating kinase 1 (ASK1) [55,56]. Therefore, downregulation of antioxidants and HSPs during usual medical treatment could result in a higher-pro-oxidant environment and decreased cellular ability to respond to stress (e.g., oxidative stress), contributing negatively to clinical outcomes.

The role of PA, specifically regular exercise training, has been attributed to increased antioxidant capacity and decreased oxidative-stress biomarkers in both healthy and diseased conditions [25,31,51,55,56,57,58,59,60,61]. For instance, data, from our and other laboratories, on trained subjects showed that the adaptive response of PBMCs in terms of modulation of antioxidants and stress-induced markers was linked to rapid and substantial changes in the gene-expression pattern, which correlates with improvements in other parameters of health status [25,31,51,54,62,63,64]. Moreover, several studies showed that regular exercise leads to an increase in antioxidant activity at the systemic level [55,57,58,59], which, in turn, has been associated with decreased proliferative activity in breast tissue [65,66].

Our data confirmed a positive impact exerted by exercise training, even though differences in experimental design make comparing our results with those already published in the exercise-oncology field difficult [55,57,58,59]. Indeed, the exercise training helped to maintain constant plasmatic levels of antioxidants and cellular gene expression of SOD1 and HSP27, as well as to induce a significant increase in SOD2 expression at the cellular level, preventing their depletion that would have been caused by medical treatment. In light of these results, it is plausible to hypothesize that this improved management of the redox-state parameters would also likely affect blood cytokine levels, reducing the risk of BC development and progression as well as reducing distress symptoms that affect QoL in BC patients.

Several biological pathways may explain the beneficial effects of PA on BC progression or recurrence, including an effect on mediators of inflammation. The mechanisms through which PA ameliorates inflammation are not yet fully clarified; however, it is known that independently from the characteristics of exercise (i.e., type, intensity, frequency, and duration), regular participation in PA has a positive effect on the immune system, so much so that it can be considered a kind of “immunotherapy” capable of reducing risk of developing inflammatory-related diseases [67,68,69,70,71]. Similarly to previous research [72,73,74], we found that a customized online and home-based PA program would promote an anti-inflammatory response. Particularly, in BC patients that undertook PA for 16 weeks, there was a significant decrease in IL6 and an increase in IL10.

IL6 is a proinflammatory cytokine released by various cells and plays a critical role in expansion and differentiation of cancer cells [75]. This cytokine exerts these functions in cooperation with various pathways; however, among them, IL6R/JAK/STAT3 is the most important player in the pathway [75]. Increased levels of IL6, both in circulation and at the tumor site, have been demonstrated in BC [76]. While this increase is usually associated with poor prognosis and lower survival in BC patients, downregulation of IL6 is related to better responses to treatment and improved QoL [77,78,79,80].

IL10 is a cytokine produced by almost all leukocytes [81], with the capacity to inhibit release of proinflammatory mediators [82]. Similarly to IL6, binding of this cytokine with its receptor (IL10R1) causes phosphorylation of JAK1 and TYK2. Once phosphorylated, JAK1 further phosphorylates a signal transducer and an activator of transcription-3 through AMPK. The signal transducer and activator of transcription-3 translocate into the nucleus and upregulate anti-inflammatory genes. Interleukin-10 also activates the PI3K/Akt/mTORC pathway and inhibits GSK3β to promote anti-inflammatory responses [83]. In BC, IL10 predominantly performs three biological activities that contribute to tumor-inhibiting action: (1) promoting CD8+ T-cell activation and proliferation and (2) inhibiting both T-cell-stimulated tumor-killing immunity by suppressing antigen presentation by APCs and (3) tumor-promoting inflammation [83,84,85]. Interestingly, IL6 is known to promote tumor growth by upregulating antiapoptotic and angiogenic proteins in tumor cells [75], and elevated IL10 may inhibit tumor growth by suppressing IL6 production. This is supported by the observation of an inverse correlation between IL10 and IL6 levels in breast cancer patients [86].

Therefore, in the complex inflammatory milieu typical of BC, our results confirm that PA could be an important strategy used alongside clinical routine to moderate the inflammatory process during medical treatments. In addition, this could impact several intracellular signaling pathways involved in cancer-cell growth, metastasis, and inflammation, as well as distress symptoms, which negatively affect the QoL of BC patients.

Quality of life is a significant issue in BC patients and survival. During treatment, about 54% of patients experience pain and sleep disturbances as well as an increase in body weight associated with reduced functional capacity and cardiorespiratory fitness [14,78,87]. PA has been shown to be effective in improving physical functioning, in reducing treatment-related fatigue, and in enhancing QoL in BC survivors and may be effective in preventing weight gain [21,22,23,24,88,89,90,91,92]. Similarly, in agreement with the aforementioned studies, a home-based PA program during medical treatments was safe and feasible and had beneficial effects on physical fitness and mental well-being. As expected, we found significant effects of exercise on BMI and body composition (i.e., FAT% and FFM%). A multitude of studies have demonstrated that women who are overweight/obese at the time of BC diagnosis are at an increased risk of cancer recurrence and death when compared with leaner women. Furthermore, some evidence suggests that women who gain weight after BC diagnosis may also be at an increased risk of poorer clinical outcomes [93]. Exercise also improved the functional capacities of patients enrolled in our study, which is in agreement with Galiano-Catillo and colleagues [21]. Indeed, the 6MWT could be used as a measure of global health in women with breast cancer [21]. Moreover, the improvement in mobility and flexibility of the upper limbs highlights how this protocol could reduce negative side effects induced by invasive surgeries (i.e., mastectomy). However, in our study, significant improvement was shown only in the physical function subscale of the EORTC-C-30 and in the cognitive fatigue scale of the FA-12. Thus, despite the fact that exercise was proven to improve QoL and fatigue, there were only slight differences in their respective domains. This is likely due to differences in the characteristics of the patients, the exercise, and the measurement methods when compared with other published results.

Limitations to this pilot study include the small sample size and the use of a heterogeneous BC population (various subtypes of BC) with various clinical characteristics (i.e., tumor size, nodal status, histological grade) and undergoing different adjuvant therapies (radiotherapy, hormone therapy, chemotherapy). This may have hidden some significant changes in the parameters we considered. Therefore, further large prospective studies are required to determine the impact of adapted PA on specific subtypes of BC, focusing on the study of their molecular/mechanistic aspects and their possible correlation with specific clinical outcomes. Moreover, follow-up should be considered to understand the duration of beneficial effects.

## 5. Conclusions

In conclusion, this pilot study highlights promising results demonstrating that in postsurgery BC patients, a 16-week online and home-based PA intervention, supervised by a kinesiologist, was not only crucial in improving functional and anthropometric parameters but may have also activated cellular responses through a multitude of actions. These may have included modulation of gene expression and protein activity, impacting several signaling pathways/biological activities involved in tumor-cell growth, metastasis, and inflammation, as well as moderating distress symptoms known to negatively affect QoL (Figure 4).

Given the massive amounts of reports linking ROSs to almost every step of tumorigenesis, various authors have proposed that ROS-generating and -scavenging systems are potential targets for cancer therapy. However, to date, less attention has been given to development of alternative redox-system-targeted strategies for BC therapy, such as PA, to complement current adjuvant therapy. We strongly believe that our findings will provide useful information for investigators who conduct exercise trials in cancer populations, clinicians who treat women diagnosed with breast cancer, and those who develop community-based exercise programs for cancer survivors.

## Figures and Tables

**Figure 1 antioxidants-12-01138-f001:**
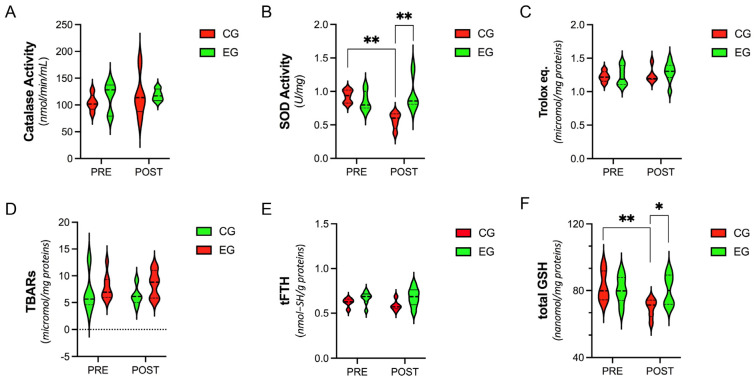
Violin−plot analysis comparing the plasma antioxidants and oxidative−stress markers measured at baseline (PRE) and at the end of the experimental period (POST) in female breast cancer patients assigned either to the Control Group (CG, *n* = 10), where all subjects received the usual cancer treatments, or to the Exercise Group (EG, *n* = 10), where they were additionally included in a 16−week online exercise training program. (**A**) Catalase and (**B**) SOD activity, (**C**) TAC and (**D**) TBARs assays, (**E**) tFTHs, and (**F**) total GSH levels. Statistical significance was determined using two−way ANOVA with Bonferroni’s post hoc analysis. * *p* < 0.05; ** *p* < 0.01. SOD, superoxide dismutase; TAC, total antioxidant capacity; TBARs, thiobarbituric acid-reactive substances; tFTHs, total free thiols; GSH, glutathione.

**Figure 2 antioxidants-12-01138-f002:**
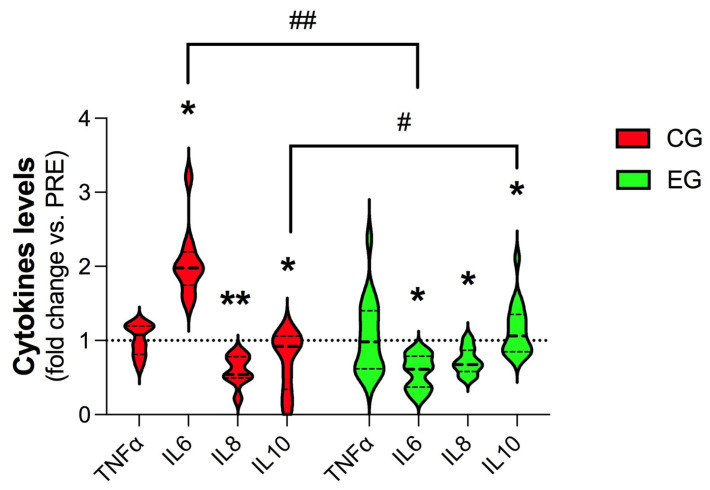
Violin-plot analysis comparing the plasma concentrations of TNFα, IL6, IL8, and IL10 measured at the baseline (PRE) and at the end of the experimental period (POST) in female breast cancer patients assigned either to the Control Group (CG, *n* = 10), where all subjects received the usual cancer treatments, or to the Exercise Group (EG, *n* = 10), where they were additionally included in a 16-week online exercise training program. Statistical significance was determined using two-way ANOVA with Bonferroni’s post hoc analysis. Each violin plot shows the fold changes relative to baseline levels. * *p* < 0.05, ** *p* < 0.01, # *p* < 0.05; ## *p* < 0.01.

**Figure 3 antioxidants-12-01138-f003:**
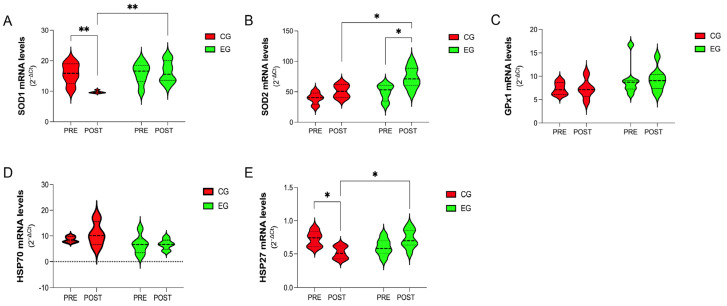
Violin-plot analysis comparing the mRNA levels of heat-shock proteins and antioxidants such as SOD1 (**A**), SOD2 (**B**), GPx1 (**C**), HSP70 (**D**), and HSP27 (**E**) in PBMCs isolated at the baseline (PRE) and at the end of the experimental period (POST) from female breast cancer patients assigned either to the Control Group (CG, *n* = 10), where all subjects received the usual cancer treatments, or to the Exercise Group (EG, *n* = 10), where they were additionally included in a 16-week online exercise training program. Statistical significance was determined using two-way ANOVA with Bonferroni’s post hoc analysis. * *p* < 0.05; ** *p* < 0.01. CG, Control Group; EG, Exercise Group; SOD, superoxide dismutase; GPx, glutathione peroxidase; HSP, heat-shock protein.

**Figure 4 antioxidants-12-01138-f004:**
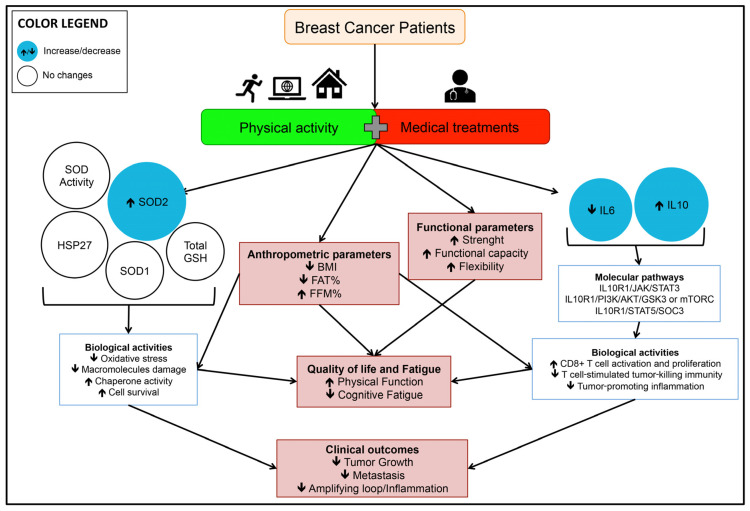
Overview of the modulation, over the experimental period, of markers/parameters measured in female breast cancer patients in the Exercise Group (EG), where patients received the usual cancer treatments and participated in a 16-week online exercise training program. These markers/parameters are important to improving functional and anthropometric parameters and to activating cellular responses through a multitude of actions (such as modulation of gene expression and protein activity that impacts several signaling pathways/biological activities) involved in tumor-cell growth, metastasis, and inflammation, as well as affecting distress symptoms known to negatively affect QoL. QoL, quality of life; SOD, superoxide dismutase; HSP, heat-shock protein; IL, interleukin; BMI, body mass index; FAT, fat mass; FFM, free fat mass.

**Table 1 antioxidants-12-01138-t001:** Characteristics of female breast cancer patients randomly assigned to the Control Group (CG), where all subjects received the usual cancer treatments, or the Exercise Group (EG), where the volunteers were additionally included in a 16-week online exercise training program.

	EG (*n* = 10)	CG (*n* = 10)	*p*-Value *
Age (Years)	50.55 ± 5.69	45.15 ± 5.54	0.874
Weight (Kg)	60.30 ± 5.55	55.30 ± 1.05	0.404
Height (m)	161.68 ± 8.61	157.51 ± 3.55	0.349
BMI (Kg/m^2^)	23.10 ± 2.53	21.38 ± 0.50	0.248
**Type of Intervention**			
Quadrantectomy	7	6	N.A.
Mastectomy	3	4	N.A.
**Treatments**			
Chemo + Hormonal + Radio	2	2	N.A.
Hormonal + Radio	4	5	N.A.
Hormonal	4	3	N.A.
**Physical Activity Level**			
IPAQ (MET-min/week)	898.3 ± 988.5	885.6 ± 666.0	0.695

Values are presented as mean ± standard deviation (SD); BMI, body mass index; IPAQ, International Physical Activity Questionnaire; MET, metabolic equivalent; min, minutes; m, meter; Kg, kilogram; N.A., not applicable. * *p*-value was determined with Student’s *t*-test of the difference between baseline characteristics of both groups (EG vs. CG).

**Table 2 antioxidants-12-01138-t002:** Functional test analysis performed at the beginning (PRE) and at the end (POST) of the experimental protocol in female breast cancer patients, randomly assigned to the Control Group (CG, *n* = 10), where all subjects received the usual cancer treatments, or the Exercise Group (EG, *n* = 10), where volunteers were additionally included in a 16-week online training program.

	PRE (Mean ± SD)	POST (Mean ± SD)	% Change	*p*-Value *
**6MWT** (m)				
EG	564.42 ± 50.78	601.1 ± 52.11	+6.50	**0.007**
CG	530.41 ± 48.78	480.1 ± 32.09	−9.43	**0.044**
**Borg** (0–10)				
EG	2.75 ± 0.43	1.12 ± 0.95	−59.18	**0.003**
CG	1.19 ± 0.28	1.74 ± 0.50	+42.00	0.172
**HGR** (Kg)				
EG	29.91 ± 3.75	28.90 ± 4.62	−3.49	0.422
CG	29.90 ± 3.69	28.90 ± 4.64	−3.87	0.138
**HGL** (Kg)				
EG	27.25 ± 3.14	26.04 ± 4.04	−4.46	0.425
CG	23.14 ± 4.07	20.10 ± 3.19	−13.50	**0.002**
**30′ STS** (*n*)				
EG	18.04 ± 4.20	22.17 ± 5.51	+22.84	0.060
CG	19.04 ± 3.18	18.13 ± 5.52	−5.21	0.355
**Sit-and-Reach** (cm)				
EG	1.77 ± 8.61	6.24 ± 8.75	+250.00	**0.004**
CG	1.71 ± 8.64	2.87 ± 3.72	+57.53	0.821
**Scratch R** (cm)				
EG	24.81 ± 10.59	18.80 ± 6.73	−24.12	**0.009**
CG	23.2 ± 10	14.1 ± 6.8	−4.32	0.731
**Scratch L** (cm)				
EG	27.50 ± 7.11	22.31 ± 7.41	−18.81	**0.005**
CG	25.4 ± 6.1	26.3 ± 8.4	−4.00	0.869
**Tandem** (s)				
EG	10 ± 0	10 ± 0	0	N.A.
CG	10 ± 0	10 ± 0	0	N.A.

EG, Exercise Group; CG, Control Group; 6MWT, 6-min walking test; HGR, handgrip right; HGL, handgrip left; STS, sit-to-stand; L, left; R, right; N.A., not applicable. All data are presented as mean ± standard deviation (SD). * *p*-value was determined with Student’s *t*-test of the difference between PRE and POST within each group. Statistically significant *p*-values are in bold. Percentage change reported was calculated as changes in mean values divided by the absolute mean values of the original values, multiplied by 100.

**Table 3 antioxidants-12-01138-t003:** Assessment of body composition in female breast cancer patients randomly assigned to the Control Group (CG, *n* = 10), where all subjects received the usual cancer treatments, or the Exercise group (EG, *n* = 10), where volunteers were additionally included in a 16-week online training program. These measurements were made at the beginning (PRE) and at the end (POST) of the experimental protocol.

	PRE (Mean ± SD)	POST (Mean ± SD)	% Change	*p*-Value *
**Weight** (kg)				
EG	60.30 ± 5.55	59.72 ± 6.81	−0.95	0.560
CG	55.30 ± 1.90	58.54 ± 4.87	+4.51	0.214
**BMI** (Kg/m^2^)				
EG	23.10 ± 2.53	22.71 ± 2.85	−1.35	0.214
CG	21.38 ± 0.50	22.74 ± 2.84	+6.55	**0.045**
**FFM** (%)				
EG	72.57 ± 5.25	74.60 ± 6.50	+2.80	**0.049**
CG	60.67 ± 5.32	60.61 ± 3.5	−0.67	0.243
**FAT** (%)				
EG	27.43 ± 5.25	25.53 ± 6.65	−6.93	**0.043**
CG	30.18 ± 4.05	34.04 ± 4.43	+12.71	**0.032**
**TBW** (%)				
EG	54.14 ± 5.63	56.23 ± 7.21	+3.86	0.072
CG	41.14 ± 5.31	40.22 ± 3.20	−2.11	0.560
**BCM** (%)				
EG	37.95 ± 3.74	38.21 ± 4.05	+0.85	0.328
CG	36.12 ± 2.17	32.41 ± 2.12	+3.86	0.124

BMI, body mass index; FFM, free fat mass; FAT, fat mass; TBW, total body water; BCM, body cell mass. All data are presented as mean ± standard deviation (SD). * *p*-value was determined with Student’s *t*-test of the difference between PRE and POST within each group. Statistically significant *p*-values are in bold. Percentage change reported was calculated as changes in mean values divided by the absolute mean values of the original values, multiplied by 100.

**Table 4 antioxidants-12-01138-t004:** Assessment of quality of life and fatigue in female breast cancer patients randomly assigned to the Control Group (CG, *n* = 10), where all subjects received the usual cancer treatments, including chemotherapy and/or hormonal therapy and/or radiotherapy, or the Exercise Group (EG, *n* = 10), where volunteers were additionally included in a 16-week online training program. These measurements were made at the beginning (PRE) and at the end (POST) of the experimental protocol.

	PRE (Mean ± SD)	POST (Mean ± SD)	% Change	*p*-Value *
**EORTC QLQ C-30**				
*Physical Function*				
EG	88.12 ± 8.01	93.32 ± 4.71	+5.78	**0.042**
CG	90.47 ± 3.55	91.52 ± 4.52	+1.06	0.345
*Emotional Function*				
EG	76.85 ± 11.62	87.03 ± 22.19	+13.23	0.242
CG	90.43 ± 8.92	89.91 ± 8.92	−0.69	0.438
*Cognitive Function*				
EG	87.03 ± 16.19	88.81 ± 14.43	+2.12	0.782
CG	90.47 ± 13.11	88.30 ± 11.30	−2.70	0.367
*Social Function*				
EG	81.48 ± 19.44	92.59 ± 14.69	+13.65	0.169
CG	88.02 ± 15.85	85.70 ± 14.99	−2.68	0.388
*Global Health*				
EG	62.96 ± 9.42	69.54 ± 15.02	+10.24	0.300
CG	72.99 ± 11.40	66.64 ± 22.22	−10.71	0.285
**FA-12**				
*Physical Fatigue*				
EG	18.51 ± 16.25	14.81 ± 14.44	−20.01	0.532
CG	12.38 ± 8.92	13.32 ± 8.60	+7.15	0.843
*Emotional Fatigue*				
EG	12.39 ± 11.75	8.6 ± 22.05	−30.32	0.707
CG	7.94 ± 13.92	9.5 ± 13.41	+16.57	0.832
*Cognitive Fatigue*				
EG	18.51 ± 21.15	7.40 ± 14.59	−60.00	**0.044**
CG	4.71 ± 12.54	7.94 ± 13.92	+40.00	0.662

All data are presented as mean ± standard deviation (SD). * *p*-value was determined with Student’s *t*-test of the difference within each experimental group. Statistically significant *p*-values are in bold. Percentage change reported was calculated as changes in mean values divided by the absolute mean values of the original values, multiplied by 100.

## Data Availability

The data presented in this study are available on request from the corresponding author. The data are not publicly available due to ethical and privacy reasons.

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
