# Peer review of "Online Home-Based Physical Activity Counteracts Changes of Redox-Status Biomarkers and Fitness Profiles during Treatment Programs in Postsurgery Female Breast Cancer Patients"

_antioxidants, 2023, doi:10.3390/antiox12051138_

Round 1

Reviewer 1 Report

ANTIOXIDANTS-2378174 presents results for a physical activity program in female breast cancer patients. While some parts of this paper were interesting, other areas could be improved. I hope the authors consider my feedback.

·         The abstract should include key results with statistics.

·         Lines 81-97: Much of this information is not necessary and arguably more so methods. Maybe just finish the Introduction with a clear purpose statement.

·         Line 117: Block randomization?

·         Methods: Details about the protocol for some of the measures could be better stated. For example, the handgrip strength test protocol is too brief. Expand as appropriate.

·         Methods: Was the design or analyses guided by key factors such as cancer stage, treatment, age, etc?

·         The statistical analysis section needs more detail. For example, what exactly was analyzed with the ANOVA?

·         Table 2, for example, does not align with an ANOVA. Were there other statistical tests such as t-tests?

·         Tables: Please list all p-values, including those that are not significant.

·         Please list throughout, including the title, that this is a pilot investigation considering design and sampling.

·         Section 6 should come before Section 5, perhaps instead as sub-section 4.1.

·         Make any changes to the abstract that align with those from the results.

Author Response

Reviewer comments:

Reviewer 1

ANTIOXIDANTS-2378174 presents results for a physical activity program in female breast cancer patients. While some parts of this paper were interesting, other areas could be improved. I hope the authors consider my feedback.

Q1. The abstract should include key results with statistics.

A1. As suggested by Reviewer, we revised the abstract including results and statistics.

Q2. Lines 81-97: Much of this information is not necessary and arguably more so methods. Maybe just finish the Introduction with a clear purpose statement.

A2. We thanks the Reviewer for this comment, however, we believe that the information reported in this part of the introduction is important for understanding the work.

Q3. Line 117: Block randomization?

A3. We thank the Reviewer for this comment. We added a section “Randomization and Blinding Procedures” explaining the procedure that we followed to randomize participants.

Q4. Methods: Details about the protocol for some of the measures could be better stated. For example, the handgrip strength test protocol is too brief. Expand as appropriate.

A4. As suggested from the Reviewer, we added more details about the muscular strength tests

Q5. Methods: Was the design or analyses guided by key factors such as cancer stage, treatment, age, etc?

A5. As we stated in the section 2.1 (Study Design), all female volunteers had diagnosed with stage I-III breast cancer, >18years old and scheduled to undergo adjuvant therapy (radiotherapy, hormone-therapy, chemotherapy). Being a preliminary study, we did not use inclusion parameters limiting to a particular type/stage of the disease. Our aims were to highlight a positive impact of a 16-week online and home-based PA intervention in post-surgery BC patients focusing on functional and anthropometric parameters, but also on cellular responses through the modulation of gene expression and protein activity, involved in several signaling pathways/biological activities. Certainly our results need to be confirmed in a larger population that might consider a specific subtype of BC and/or a particular stage of the disease.

Q6. The statistical analysis section needs more detail. For example, what exactly was analyzed with the ANOVA?

A6. We thank the Reviewer for this comment. We revised the “Statistical analysis” section specifying which data were analysed with ANOVA or with Student’s t-test.

Q7. Table 2, for example, does not align with an ANOVA. Were there other statistical tests such as t-tests?

A7. As we stated below the table 2, P-value was determined by Student’s t-test of the difference between PRE and POST within each group.

Q8. Tables: Please list all p-values, including those that are not significant.

A8. As suggested by Reviewer, we included all p-values.

Q9. Please list throughout, including the title, that this is a pilot investigation considering design and sampling.

A9. We thank the Reviewer for this comment, and to clarify the characteristic of our study we listed throughout the manuscript that it is a pilot study.

Q10. Section 6 should come before Section 5, perhaps instead as sub-section 4.1.

A10. A suggested from the Reviewer 2, we have already included the section 6 at the end of the “Discussion” section in this revised manuscript.

Q11. Make any changes to the abstract that align with those from the results.

A11. As suggested, we revised the abstract

Reviewer 2 Report

Dear Authors

Your paper is really excellent experimental study for investigate the online home-based physical activity counteracts changes of redox-status biomarkers and fitness profile during treatment programs in post-surgery female breast cancer patients. This issue is surely uncovered in sports science and medicine filed. I would like to thank the authors for their work on this manuscript.

Minor concerns                 

Abstract

Moreover, to assess exercise-induced modulatory effects, the gene expression of heat shock proteins 70 and 27 (HSP70 and HSP27), superoxide dismutase 1 and 2 (SOD1 and SOD2), and glutathione peroxidase 1 (GPx1) in peripheral blood mononuclear cells (PBMCs) was analyzed.

You should change to “Moreover, to assess exercise-induced modulatory effects, the gene expression of heat shock proteins 70 and 27, superoxide dismutase 1 and 2, and glutathione peroxidase 1 in peripheral blood mononuclear cells was analyzed.”

Introduction

In general, the Introduction section is quite good.

Method

In general, the Methods section is quite good, too.

The Ref 32 (Lee et al.’s study) is not valid and reliable study of International Physical Activity Questionnaire. Please change to proper references.

Results.

Line 318, 23.1±2.5 Kg/m2 vs. 21.3±0.5 Kg/m2 -> 23.1±2.5 Kg/m2 vs. 21.3±0.5 Kg/m2

Please insert exact p-value of each variable in Table 1 for show homogenous.

In all Tables, please change from “Mean ± SD” to “Mean ± standard deviation”.

In all Tables, please all results (data) should be displayed to two decimal places.

Discussion

The “6. Limitations” section should be moved to end of Discussion section.

I recommend that this manuscript should be edited by an English professional editor for more readable. There are several grammatical errors.

Author Response

Reviewer 2 

Minor concerns                 

Abstract

Q1. Moreover, to assess exercise-induced modulatory effects, the gene expression of heat shock proteins 70 and 27 (HSP70 and HSP27), superoxide dismutase 1 and 2 (SOD1 and SOD2), and glutathione peroxidase 1 (GPx1) in peripheral blood mononuclear cells (PBMCs) was analyzed.

You should change to “Moreover, to assess exercise-induced modulatory effects, the gene expression of heat shock proteins 70 and 27, superoxide dismutase 1 and 2, and glutathione peroxidase 1 in peripheral blood mononuclear cells was analyzed.”

A1. We revised the abstract.

Introduction

Q2. In general, the Introduction section is quite good.

A2. Thank you for this comment

Method

In general, the Methods section is quite good, too.

Q3. The Ref 32 (Lee et al.’s study) is not valid and reliable study of International Physical Activity Questionnaire. Please change to proper references.

A3. We changed the reference with the following:

Lee, P.H.; Macfarlane, D.J.; Lam, T.H.; Stewart, S.M. Validity of the International Physical Activity Questionnaire Short Form (IPAQ-SF): a systematic review. Int J Behav Nutr Phys Act. 2011; 8:115. doi: 10.1186/1479-5868-8-115.

Results.

Q4. Line 318, 23.1±2.5 Kg/m2 vs. 21.3±0.5 Kg/m2 -> 23.1±2.5 Kg/m2 vs. 21.3±0.5 Kg/m2

A4. We revised the text.

Q5. Please insert exact p-value of each variable in Table 1 for show homogenous.

A5. The p-value of each variable was included in Table 1.

Q6. In all Tables, please change from “Mean ± SD” to “Mean ± standard deviation”.

A6. We thank the Reviewer for this comment, and to keep the format of the tables not too big we have specified in the notes under each table that the reported values indicate the mean±standard deviation (SD).

Q7. In all Tables, please all results (data) should be displayed to two decimal places.

A7. All tables have been revised.

Discussion

Q8. The “6. Limitations” section should be moved to end of Discussion section.

A8. As suggested by Reviewer, we moved the “Limitations” to the end of “Discussion” section.

Comments on the Quality of English Language

Q9. I recommend that this manuscript should be edited by an English professional editor for more readable. There are several grammatical errors.

A9. We regret there were problems with the English. Our paper has been carefully revised by a native English speaker to improve the grammar and readability.

Round 2

Reviewer 1 Report

The authors have done a fair job addressing my previous concerns. One last point to further strengthen the paper:

***Tables 2, 3, 4: Change the columns in these tables from delta pre-post % to delta mean+-SD. 

Author Response

Reviewer 1

The authors have done a fair job addressing my previous concerns. One last point to further strengthen the paper:

Q1.***Tables 2, 3, 4: Change the columns in these tables from delta pre-post % to delta mean+-SD. 

A1. We tank the Reviewer for this comment. The title of the column was probably misleading and does not reflect what we intended to indicate in the table. In particular, the values shown refer to the percentage of changes (% Change) of each parameter calculated as changes in mean values divided by the absolute mean values of the original values, multiplied by 100. % Change=((POST-PRE)/PRE)*100. To clarify this important aspect, we changed the title of the column and added a sentence below the tables. With this column we want to give the reader more information about the trend of the specific parameter, regardless of the statistical data obtained on the raw values.